# From General Circulation to Global Change: The Evolution, Achievements, and Influences of Duzheng Ye's Scientific Research

Jianhua Lu

School of Atmospheric Sciences, Southern Marine Science and Engineering Guangdong Laboratory (Zhuhai), Sun Yat-sen University (SYSU), Zhuhai 519082, China; lvjianhua@mail.sysu.edu.cn

**Abstract:** Duzheng Ye (Tu-cheng Yeh) was an active member of Rossby's Chicago School, one of the founders of modern meteorology in China since the 1950s, and a pioneer of global change science in China and over the world. His achievements have been central to the development of atmospheric and climate dynamics and global change studies in China, and many of them remain to be fundamental in the context of global climate change. In this review, his lifelong research career is divided into five periods: (1) the preparatory period (1935–1944); (2) the Chicago period (1945–1950); (3) the period of breaking ground (1950–1966); (4) the period of transition (1972–1983); and (5) the period of global change (1984–2013). The evolution of Yeh's main achievements is described in the context of the historical background of both China and the world. These well-known achievements included the theory of energy dispersion in the atmosphere, the general circulation of the atmosphere (GCA) over East Asia and the globe, Qinghai–Tibetan Plateau meteorology, the scale-dependence theory of geostrophic adaptation (adjustment), and his pioneering ideas on global change. Special emphases are put on some of Yeh's investigations that were well ahead of his time, such as his investigations on trade inversion, the GCA as an internally consistent whole, abrupt seasonal changes in the GCA, the physical mechanism of atmospheric blocking, and orderly human activities.

**Keywords:** Duzheng Ye; energy dispersion; trade inversion; the general circulation of the atmosphere; Qinghai–Tibetan Plateau meteorology; geostrophic adjustment; atmospheric blocking; global change; orderly human activities

## 1. Introduction

Professor Duzheng Ye (21 February 1916–16 October 2013; also spelled as Tu-cheng Yeh, and Yeh shall be used hereafter) was one of the active members in Rossby's Chicago School before his return to China and one of the founders of modern meteorology in China since the 1950s. He was also one of the pioneers who initiated and steered international global change research in the early 1980s [1–4].

The late Prof. Shiyan Tao, a lifelong colleague of Yeh's since 1950, together with Prof. Zhongxiang Hong, briefly summarized Yeh's achievements from three perspectives: (1) achievements in research and training; (2) the establishment of the Institute of Atmospheric Physics (IAP) in the Chinese Academy of Sciences (CAS); (3) coordinating international cooperation [5]. In terms of achievements in research and training, they mentioned energy dispersion, the adaptation (adjustment) of atmospheric motions, atmospheric general circulation, Tibetan Plateau meteorology, and climate dynamics and global change. This review addresses Yeh's achievements in research only.

Although Yeh's many accomplishments are well-known by scientific communities and his cause is developing well both in China and worldwide, there is still a need to provide a relatively thorough retrospective on the evolution of his scientific ideas and investigations via a meta-analysis of his published works. The necessity comes from the fact that many of Yeh's investigations were highly prospective, with some of them well ahead of his time,

and hence the aim of this review is not so much to honor the past achievements of a giant in atmospheric sciences, but to inspire new scientific findings from Yeh's previous works.

As an esteemed scientist, Yeh never put himself in a lofty position. On the contrary, he always considered that these accomplishments were not just his, but also belonged to his colleagues and the meteorological and climatic communities at large in China. He was most proud not of the numerous prizes and honors he obtained, but that he could dedicate his life and talent with his scientific endeavors to the advancement of science in China and the civilization of the world. Therefore, it is appropriate to place this review in the broader context of the historical background of both China and the world.

The organization of the review follows chronologically the five periods in Yeh's lifelong research career, which are named by the author of the review as follows: (I) the preparatory period (1935–1944) in Section 2; (II) the Chicago period (1945–1950) in Section 3; (III) the period of breaking ground (1950–1966) in Section 4; (IV) the period of transition (1972–1983) in Section 5; (V) the period of global change (1984–2013) in Section 6. Each of the main fields to which Yeh contributed will be described in one of the five periods for the sake of narrative clarity. Section 7 is a brief summary.

## 2. Period I: The Preparatory Period (1935–1944)

During this period, Yeh initially enrolled as a Physics student at Tsinghua University in 1935; then, he shifted his interest to meteorology in 1938 due to its crucial role in the war against Japan's invasion of China. In 1940, Yeh successfully graduated from the Southwestern Associated University in Kunming, a united institution comprising Peking University, Tsinghua University, and Nankai University. After a very short period of teaching at a high school, Yeh became a graduate student of meteorology at Zhejiang University, which moved westward to Guizhou Province in southwestern China during the war. During this time, Yeh studied atmospheric electricity under the guidance of Changwang Tu (1902–1962), a renowned meteorologist and central figure of modern meteorological research and operational systems, and Ganchang Wang (1907–1998), a former student of Lise Meitner and a pioneer of China's nuclear science. Meanwhile, Yeh published in 1942 his first research paper on isentropic analysis (Figure 1).

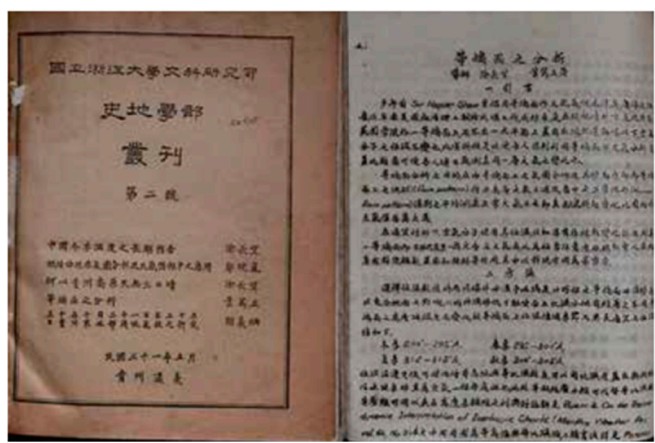

**Figure 1.** A picture of Yeh's first paper, "Analysis on Isentropic Surfaces" in *Memoir of the Division of History and Geography of Zhejiang University*, No. 2, 1942.

After completing his studies at Zhejiang University in the summer of 1943, Yeh joined the Meteorological Institute, which was one of the first eight institutes in Academia Sinica established in 1928, situated in Beibei, Chongqing during the war. Yeh served as a research assistant there until his arriving at the University of Chicago in early 1945 as a doctoral student under the supervision of C.-G. Rossby.

During this preparatory period, Yeh underwent comprehensive training in physics and meteorology under the guidance of China's finest mentors. His strong background

in scientific education would play a pivotal role in shaping his subsequent career and achievements.

## 3. Period II: The Chicago Period (1945–1950)

The Chicago School established by C.-G. Rossby, since his move to the University of Chicago in 1940, was undoubtedly at the center of the explosive development of modern meteorology after World War II. During this period, J. G. Charney, who had just finished his Ph.D. thesis at UCLA and was to develop his quasi-geostrophic theory and baroclinic instability theory and lead the Princeton project of numerical weather prediction, had stayed in Chicago from 1946 to 47 and lectured hydrodynamics there [6]. E. Palmén, during his extended stay in Chicago, engaged in stimulating debates with V. Starr on the respective roles of the meridional Hadley circulation and transient eddies in momentum and energy transport [7]. Prominent European meteorologists such as Tor Bergeron, Alf Nyberg, and Halvor Solberg visited Chicago and sometimes taught classes [6]. Rossby, while being occupied with his organizing duties [8], advanced his theory of the westerly jet based on the mixing of vorticity [9]. He was also working on theoretical explanations for the general circulation of the atmosphere (referred to as the GCA hereafter). David Fultz was conducting his noteworthy dishpan experiment; during this period, Herbert Riehl, returning from his work at the Institute of Tropical Meteorology located in Puerto Rico, taught tropical meteorology in Chicago [7]. It was during these "great days of the Chicago School" [8] that Yeh came to the very center of world meteorology and, as an active member of the Chicago School, soon contributed to the explosive development of the field. Indeed, this may also be considered as the formative period of Yeh's entire research career.

### 3.1. On Energy Dispersion in the Atmosphere

Rossby initially introduced the concept of group velocity into meteorology [10] in 1945, the same year that Yeh arrived at Chicago, and Yeh chose the topic of energy dispersion in the atmosphere as his doctoral thesis. Before his graduation in 1948, part of his results had been included in the seminal 1947paper, "On the General Circulation of the Atmosphere in Middle Latitudes", which was led by Rossby and Palmén, with Yeh being one of the participants [11] (note: the reference numbers in bold indicate that Yeh was the author or one of the authors of the publication). Figure 23 of [11], which was also included in Yeh's "energy dispersion" paper published one and a half years later (Figure 9 in [12]), illustrated the establishing process of a one-dimensional stationary wave as a result of the injection of cyclonic vorticity (wave source) at a prescribed longitude, based on the theoretical calculation in [12]. More than thirty years later, Brian Hoskins and David Karoly developed a theory for stationary Rossby wave propagation in a spherical atmosphere [13], which can be considered a two-dimensional extension of Yeh's one-dimensional theory. Interested readers may refer to [14,15] for a better understanding of the role of energy dispersion theory in atmospheric and climatic dynamics.

Indeed, the immediate and long-term applications of Yeh's energy dispersion theory went beyond this in stationary wave dynamics. It is interesting to consider Rossby's evaluation of Yeh's thesis, which was submitted to the *Journal of Meteorology* on 27 April 1948, in his letter to George Platzman (Figure 2) on 29 October 1948:

> "*Who has gone through the analytical part of Yeh's mathematics? Personally, I preferred to have his paper devoted as much as possible to basic questions, relatively less attention to detailed mathematical computations. The breakdown, dispersion, of a solitary ridge, is of importance. The coastal effects should not be included. Please let Yeh understand that for his own sake, overloading of the paper must be avoided. It will merely result in nobody reading the paper. It must be recognized that Yeh's thesis deals with the very heart of the Princeton project and must be written so as to promote this development. Best wishes, C. G. R.*"

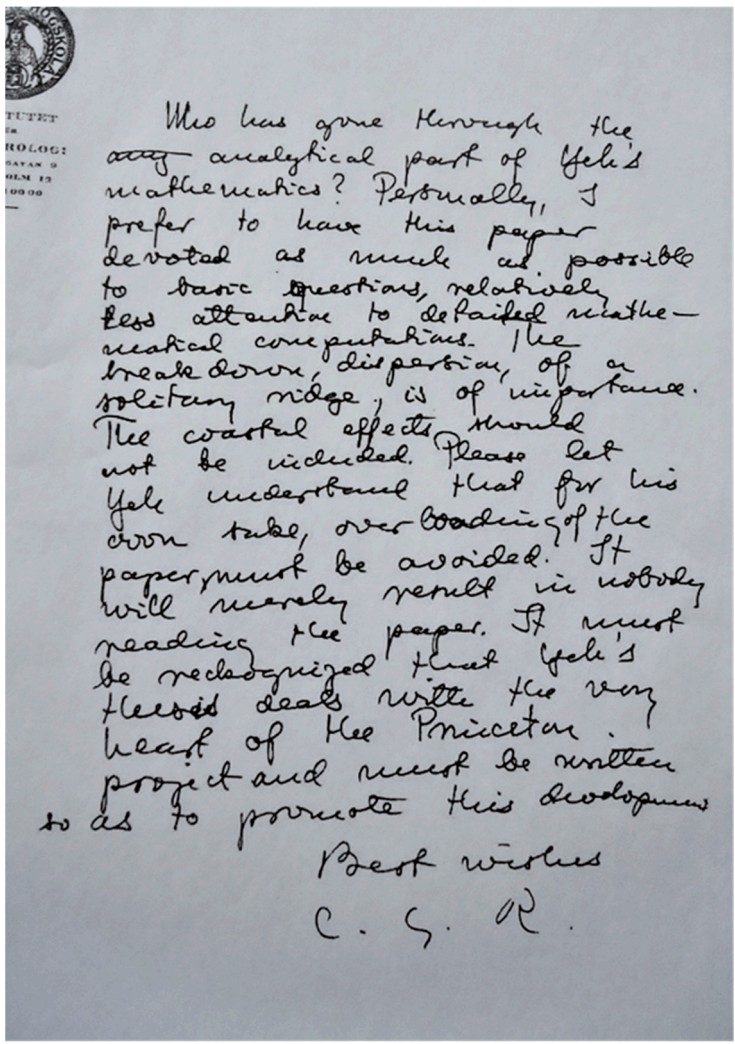

**Figure 2.** The letter of C.-G. Rossby to G. W. Platzman regarding Yeh's "energy dispersion" thesis. The date of the letter was 29 October 1948. (Photo courtesy of Anders Persson.)

'The Princeton project' in Rossby's letter refers to the numerical weather prediction (NWP) project led by Charney and von Neumann [16]. The reason why "Yeh's thesis deals with the very heart of the Princeton project" was because the results in Yeh's thesis were essential to Charney's choice of a minimum forecast area that could ensure the success of the NWP project. N. A. Phillips, who was one of the team members of the project, later further stated that if too small an area had been chosen, the failure of the NWP project would have led to a 'tremendous setback' for Charney's quasi-geostrophic theory [17] (p. 4). Indeed, Rossby not only asked Charney to write a brief note to accompany Yeh's paper [18] (p. 126), but also wrote by himself another note in the first issue of the first volume of *Tellus*, which was launched by Rossby in Sweden, to introduce Yeh's paper to a broader audience [19].

The importance of "the breakdown, dispersion, of a solitary ridge" that Rossby emphasized in his letter refers to Yeh's theoretical analyses and calculations on the different dispersive features of solitary waves at low, middle, and high latitudes (see Section 5 and Figures 10–12 in [12]), which were used to understand fast dispersion at low latitudes and the preferred occurrence and maintenance of blocking highs over high latitudes, as was stated by Yeh himself in [12].

The abstract and main contents of [12] indicated that Yeh understand the potential applications of his work well. The first application, "the formation of a new trough over North America following an intense cyclogenesis in the Gulf of Alaska" [12], was soon

visualized by E. Hovmöller in his troughs-and-ridges diagram [20] and was used to explain recent extreme cold events [21]. In the second application, Yeh explained the formation of blocking action at high latitudes by the dispersion of an initial solitary wave, in contrast to Rossby's earlier hypothesis based on the convergence of group velocity [10]. Later, during the late 1950s and early 1960s, Yeh and his Chinese colleagues would conduct a systematic survey and series of theoretical and numerical studies on blocking highs (see Section 4.4).

### 3.2. On Hadley Circulation, Trade Inversion, and the Motion of Tropical Storms

Yeh continued his pursuit of the general circulation over middle latitudes during the period 1949–1950 after the energy dispersion paper was published. For example, he discussed the maintenance of zonal circulation from the perspective of vertical and meridional vorticity transport [22]. During this period (1949–1950), while Rossby spent most of his time at Stockholm, Yeh joined Riehl's group on tropical meteorology, in which Yeh was second to Riehl [7]. Together with Riehl and the other team members, Yeh made significant contributions to the field of tropical meteorology at its very beginning. One of these contributions was the proof of the existence of the Hadley circulation. To settle the debate between Palmén and Starr on the role of the Hadley circulation in the global circulation system [7], Riehl and Yeh proved for the first time, by computations based on data obtained from the climatic atlas, the existence of meridional Hadley circulation [23].

Following their work on the Hadley cell, they further explored the spatial structure and the processes governing the balance of radiation, surface sensible and latent heat, and momentum in the north-east trade wind of the Pacific Ocean. The resultant 1951 publication by Riehl, Yeh, J. Malkus (later J. Simpson), and N. La Seur [24] is widely considered a seminal paper on the physics of tropical trade inversion and is one of the most celebrated contributions to meteorology of the twentieth century [7]. Interested readers may find a detailed observational and historical background for [24] in [7]. By carefully reading the 29-page paper containing 24 figures, one cannot help feeling the comprehensive and physically penetrating nature of the analyses that Riehl, Yeh, and their colleagues conducted on the observations obtained from the five weather ships. They concluded that the trade inversion is not a discontinuity separating the upper-layer dry air and lower-layer moist air, and that downward mass transfer occurs through the inversion. Their conclusions inspired the later development of theoretical and numerical models of trade cumuli, which are essential to the parameterization of shallow convections in climate models [25,26]. Indeed, in the mind of the author of this review, the broad view presented by Riehl, Yeh, and their colleagues in [24] may well help provide a better perspective on the relations between clouds, circulation, the hydrological cycle, and global climate change.

Before Yeh's return to China in 1950, he also conducted an investigation on the mechanism leading to the oscillating trace of tropical storms [27] and a survey on the rainfall over the islands of Hawaii [28], which originated from the collaborative research between Rossby's School and the Pineapple Research Institute in Hawaii [8].

### 4. Period III: The Period of Breaking Ground (1950–1966)

The education and research in modern meteorology in China was relatively limited in terms of scope and depth before 1949 [29]. The situation underwent a significant transformation after the establishment of the People's Republic of China. The former Institute of Meteorology in Academia Sinica was reorganized as the Institute of Geophysics and Meteorology (IGM) under the Chinese Academy of Sciences (CAS) in 1950, which was the main body of meteorological and geophysical studies in China until 1966, when it was separated into several institutes. Yeh and Chen-chao Koo [30], both protégés of Rossby who returned to China in 1950 (Koo from Stockholm), were the two directors responsible for leading the meteorological studies in the institute, under the leadership of their former teacher, J. J. Jaw, who was the director-general of the IGM.

Indeed, in 1973, two renowned atmospheric dynamicists from the U.S., W. Blumen and W. M. Washingtion, noted that, based on translated meteorological publications from

China, there was 'accumulated evidence' suggesting that "the field of meteorology (in China) had become a well established and continually growing scientific activity" between 1949 and 1966 [31]. They acknowledged the need for "a more exhaustive overview of the contributions made by Chinese meteorologists to the theory of the general circulation" [31]. It is worth mentioning that the groundbreaking works undertaken by Yeh and his colleagues at the IGM and in the universities played a pivotal role in this remarkable development. It was their efforts that realized Rossby's expectations [32] of an "extremely vigorous development of Chinese meteorology" and "many significant realistic contributions" in a relatively short span of approximately 16 years.

*4.1. On the General Circulation over East Asia*

Even prior to and, presumably, for his return to China, Yeh paid attention to the GCA over China and East Asia and published in *Tellus* a paper titled "The circulation of the high troposphere over China in the winter of 1945–46" [33]. Yeh unveiled, for the first time, the profound influence of the Qinghai–Tibetan Plateau on the upper tropospheric circulation, utilizing the best available data at that time. Within this seminal paper, Yeh observed the abrupt appearance of a new westerly jet at the southern flank of the Qinghai–Tibetan Plateau during mid-October, which merged downstream with the northern westerly jet into one singular and robust jet stream extending beyond the edge of the continent [33].

Given the pivotal importance of comprehending the general circulation over East Asia, which remained predominantly unexplored in 1950 both within China and worldwide, Yeh and his colleagues (Figure 3) in the IGM embarked on a systematical investigation of its three-dimensional structures, seasonal cycle, associated synoptic phenomena, and underlying principles. This dedicated pursuit began with the establishment of the Section of Synoptic and Dynamic Meteorology in the institute in 1950. A series of high-quality papers were published, most of them in Chinese and some in English, during this groundbreaking period.

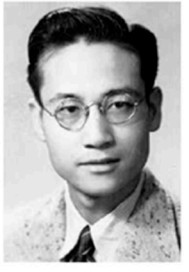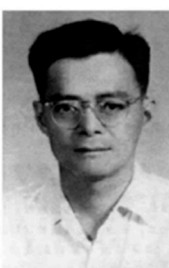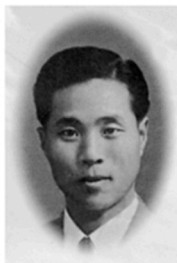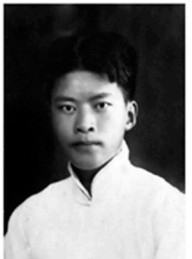

Duzheng Ye
(Tu-Cheng Yeh)
(1916-2013)

Zhenchao Gu
(Chen-chao Koo)
(1920-1976)

Shiyan Tao
(Shih-yen Dao)
(1919-2012)

Jianchu Yang
(Chien-chu Yang)
(1915-1990)

**Figure 3.** Photos of Ye–Gu–Tao–Yang (Yeh–Koo–Dao–Yang), a legendary team leading the meteorological research in the IGM, under the leadership of J.J. Jaw, the director-general of the institute, during the period 1950–1966. Yeh and Koo were the two directors of the meteorological section in the IGM. (Photos courtesy of the family members of Ye, Gu, Tao, and Yang.)

A synthesis of their findings was initially presented by Chen-chao Koo to the international meteorological community on the evening of 4 June 1957, during the Numerical Weather Prediction Conference in Stockholm, through a seminar organized to showcase the meteorological achievements in China [34]. The audience of the seminar comprised C.-G. Rossby, J. G. Charney, B. Bolin, E. N. Lorenz, N. A. Phillips, and A. M. Obukhov, among others. Parts I and II of a manuscript titled "On the general circulation over Eastern Asia" [35,36] were submitted to *Tellus* on the following day of the seminar, with Part III [37] being submitted four months later. In a groundbreaking move, Yeh and his colleagues presented the international meteorological community with a systematic and physically coherent depiction of the GCA and weather systems over East Asia. This included elucidating

their three-dimensional structures, seasonal disparities, oft-abrupt seasonal transitions, the development of associated weather systems, and the discernable influence of the Qinghai–Tibetan Plateau on circulation patterns and weather systems. These features were vividly illustrated through the 27 meticulously crafted figures presented in Part I and II of the three *Tellus* papers [35,36].

Yeh and his colleagues did not limit their attention to merely descriptive studies, but sharply realized fundamental theoretical problems while analyzing the phenomenological aspects of the general circulation over East Asia. These theoretical problems included, but were not limited to, the thermal and dynamical effects of huge orographic features on the general circulation, the role of the heat sources and sinks associated with the land–sea distributions, the role of orography in determining the positioning of mean troughs and ridges (now referred to as stationary waves), and the mechanism underlying abrupt seasonal changes in the general circulation. For instance, Figures 3 and 4 in Part III [37] of the three *Tellus* papers, obtained from a calculation applying the Green's function method to a steady, linearized Sawyer–Bushby model [38] forced by the topography of the Qinghai–Tibetan Plateau and the distribution of heat sources and sinks over the Northern Hemisphere, distinctively illustrated the respective roles of the topography and diabatic heating in the formation of a deep wintertime trough along the East Asian coast.

### 4.2. On the Meteorology of the Qinghai–Tibetan Plateau

While previous research had predominantly focused on the mechanical influence of topography on atmospheric circulation [39], Yeh and his colleague C.-C. Koo were quick to recognize and emphasize, as early as 1955, the potential thermal impact of large-scale topographic features, such as the Qinghai–Tibetan Plateau [40]. Through meticulous calculations using available aerological and surface observations from 1955–1956, Yeh and his two associates unequivocally stated in 1957 that the Qinghai–Tibetan Plateau acts as a heat source during summer and likely a heat sink (except in the southeastern region of the plateau) during winter [41]. This pioneering study stands as the first reliable calculation of the thermal effect of the Qinghai–Tibetan Plateau. Subsequently, Yeh and his colleagues devoted themselves to studying various aspects of the Qinghai–Tibetan Plateau's role in the GCA, climate, and weather, not only over China and East Asia, but also over the entire globe. Their research ultimately led to the establishment of the field known as "the Meteorology of the Tibetan Plateau" [42] or "the Meteorology of the Qinghai–Tibetan Plateau" [43], as suggested by the two monographs published in 1960 and 1979.

The investigation of Qinghai–Tibetan Plateau Meteorology permeated Yeh's lifelong career, and he made many other contributions to this field. For the sake of narrative clarity, I will briefly outline here his significant later contributions in this area, so as to focus in the subsequent sections on the new fields he pioneered and influenced. Yeh himself summarized the studies conducted by himself and his colleagues in this field in English-language journals such as the *Bulletin of the American Meteorological Society* (BAMS) in 1981 [44] and *Meteorology and Atmospheric Physics* in 1998 [45]. These review papers provide in-depth insights into their research, and interested readers are encouraged to refer to them for further details.

During the mid-1970s, Yeh and his colleagues conducted experiments by constructing a rotating annulus with a heated "plateau" to investigate the thermal effects of the Tibetan Plateau on the general circulation. Their research focused on various aspects such as the vertical structures of horizontal circulation, the zonal and meridional overturning circulations, the convective activities, and the movements of the Tibetan High over the Plateau, as is well summarized in [46]. In 1979, Yeh and his colleague Youxi Gao led the publication of [43], which provided a comprehensive summary of the achievements in this field by the Chinese meteorological community up to that time.

During the 1980s, Yeh, along with his colleagues and students, expanded his research to explore the influences of the Qinghai–Tibetan Plateau on the regions beyond East Asia [45]. Their work became particularly significant for current studies on polar climate change

and the global effects of the Tibetan Plateau. In her doctoral thesis supervised by Yeh and G. X. Wu, X. L. Zou discovered that wintertime planetary waves with low wavenumbers (wave 1 and wave 2), induced by the mechanical forcing of the Tibetan Plateau (but not the Rocky Mountains), could propagate northward into high latitudes and vertically into the stratosphere [47,48]. This finding implied that the Tibetan Plateau plays a crucial role in shaping the spatial structure of the climate over the high latitudes of the Northern Hemisphere, affecting both its mean state and variations.

### 4.3. On the Scale-Dependence Theory of Geostrophic Adjustment

Geostrophic adjustment, known as "geostrophic adaptation" in China, has long been an important theoretical issue in atmospheric and oceanic dynamics [49]. In contrast to traditional wisdom, both Rossby in the 1930s [49] and Obukhov in 1949 [50] demonstrated that during the process of geostrophic adjustment, the mass (pressure) field rapidly aligns itself with the velocity field until the geostrophic balance is achieved. In 1957, Yeh was the first to highlight that the direction of mutual adjustment between the velocity and mass fields depends on the spatial scale of the initial geostrophic imbalance, as explicitly stated in [51]:

*"From the foregoing discussions we may give the following statement about the production of quasigeostrophic motion: When due to some reason or other the quasigeostrophic equilibrium breaks down, then for small scale motion (not so small that the earth's rotation may be neglected) it is the pressure field to fit the new velocity field to attain new quasigeostrophic motion; for very large scale it is the velocity field which changes more to give new quasigeostrophic motion; and for intermediate scale both fields will change."*

Later, Q. C. Zeng (T.-S. Tseng) mathematically proved that the initial scale of a non-geostrophic disturbance determining the direction of geostrophic adjustment (adaptation) between the mass and velocity fields is the Rossby radius of deformation [52,53]. Yeh's research served as inspiration for Chinese scientists, including some who worked directly under Yeh's influences, leading to a uniquely systematic investigation of this topic. The findings of their research were summarized in a 1965 monograph titled "The Problems of Adaptation in Atmospheric Motion", authored by Yeh and his student M.-T. Li [54]. In this monograph, they also explored the adjustment (adaptation) in meso-scale atmospheric motion and the hydrostatic adjustment.

### 4.4. On the Dynamics of Atmospheric Blocking Highs

The slower dispersion of solitary waves at high latitudes, as described in the fifth section of Yeh's energy dispersion thesis in 1949 [12], remains a fundamental theoretical paradigm for the dynamics of blocking highs, a concept that continues to be supported by recent studies [55,56]. In the late 1950s and early 1960s, Yeh and his colleagues systematically investigated the climatology, synoptic features, numerical simulations, and dynamics of atmospheric blocking over the Northern Hemisphere, based on a thorough survey of 85 wintertime blocking events (54 events over the region from the North Atlantic to the Ural Mountains and 31 events over the North Pacific) that occurred between 1955 and 1960. Two monographs were published in 1962 and 1963, respectively: one focusing on synoptic and dynamical investigations [57], and the other on numerical simulations [58]. From their survey and simulations, Yeh and his colleagues developed synoptic and conceptual models for the onset, maintenance, and decay of blocking highs. This systematic investigation into blocking highs was truly exceptional, as it took more than twenty years for another comprehensive volume on the topic to be published internationally [59].

Yeh further discussed a possible physical mechanism for the onset and decay of blocking highs. He emphasized the significant role played by the baroclinic instability of long waves (approximately 5000 km or above in scale) in the onset of the meridional-type circulation. Furthermore, he highlighted the importance of local conditions favoring the appearance of highly non-geostrophic conditions in the cut-off process of a closed high cell. Additionally, positive vorticity advection toward the blocking high leads to its decay.

Furthermore, in an article published in 1963, Yeh explicitly emphasized the essential role of nonlinear wave–wave interaction in the formation of blocking highs, as explicitly stated in the abstract of [60]:

> "…we shall keep in the vorticity and thermodynamic equations the nonlinear terms which are dropped generally. This enables us to study the mutual interaction of the disturbances. It will be shown theoretically that this mutual interaction is very important in the formation of Ω-shaped blocking highs."

### 4.5. On the Fundamental Problems of Global Atmospheric Circulation

Yeh's theoretical interest in the GCA was not limited to that over East Asia, but he also considered a more fundamental problem, i.e., the very nature of the global atmospheric circulation as a whole. While the debates and investigations within Rossby's Chicago School, in which Yeh actively participated [7,23], influenced his early interest and perspective on this matter, Yeh and his colleagues in China soon independently developed their own views through systematic research. Due to space limitations, I will only highlight two examples of his work: his monograph on some fundamental problems of the GCA [61] and his investigation on abrupt seasonal changes in the GCA [62].

#### 4.5.1. On an Internally Consistent Picture of the General Circulation

From July to September 1957, only about one year after N. A. Phillips published his seminal numerical experiment on the GCA [63], Yeh and his colleague Pao-chen Chu conducted a series of seminars in Beijing. The content of these seminars revolved around their initial draft on some fundamental problems of the GCA. Just one year later in 1958, approximately nine years before E. N. Lorenz published his monograph on the nature and theory of the GCA [64], Yeh and Chu's monograph (a 159-page volume) was published. Although the text was in Chinese, it included at the end of the book 16 pages of chapter abstracts in English [61] (p. 144–159). B. Hoskins commented, "The chapter headings give an idea of the broad sweep and ground-breaking nature of his (Yeh's) ideas at this time" [1], and the detailed chapter and section headings can be found in Figure 1 of [65]. A comparison between Yeh and Chu's classic monograph and Lorenz's indicates the different styles and visions of the three authors regarding the GCA [66], while both books demonstrated a penetrating depth of physical thinking.

In the final chapter of [61], Yeh and Chu made an effort to present an internally consistent picture of the GCA. They emphasized that the GCA is a coherent system in which various components and physical processes, influenced by external factors such as radiation and the Earth's rotation, are interconnected. Particularly, they emphasized the central role of large-scale eddies in connecting these fundamental elements of, and key physical processes in, the GCA (Figure 4, excerpts from Chapter XI). They analyzed how and why the large-scale-eddy-induced angular momentum transport in the upper layer and heat transport in the lower layer of the atmosphere should be considered an integrated entity, and their analyses were qualitatively consistent with the Eliassen–Palm flux obtained from the theory of wave–mean flow interaction by Andrews and McIntyre [67,68], which was developed during the mid-1970s.

> *In this chapter **an attempt is made to give an internally consistent picture of the general circulation.** The way to reach this goal is to try to relate the main physical mechanism with the physical processes operating in the general circulation. The following is the main discussion of this chapter.*
>
> *The basic state of general circulation may be described as follows: In the mean meridional plane there are three cells, two direct and one indirect. (This is what we call mean meridional circulation). In the horizontal plane there are planetary wind belts (i.e. westerlies and easterlies). The wind distribution is not uniform. There is the so-called jet stream. Superimposed on these wind belts are lows and highs, troughs and ridges (large-scale disturbances). We call these the basic elements of the general circulation. **They are not independent. They are mutually related and form an internally consistent integrity. In the course of formation of this integrity, … large-scale disturbances play a basic role.***
>
> *From time to time the large-scale disturbances transport and redistribute the physical properties (as heat, angular momentum etc.) of the atmosphere. **Through these transports and redistributions the large-scale disturbances tie up the elements of the general circulation.** …*
>
> *……*
>
> *In the formation of jet stream large-scale disturbances are also important.*
>
> *……*
>
> *Not only the main elements of general circulation are mutually constrained. The main physical processes of the general circulation are also mutually related. By main physical processes we mean those processes that are operating in the balance of important physical properties of the atmosphere, such as heat, water vapour, kinetic energy and angular momentum. ……From the above discussion we see that the main physical processes operating in the atmosphere are connected with one another. **In connecting these physical processes the unstable disturbances play an important role.***
>
> *……*

**Figure 4.** Excerpts from the chapter abstract, originally in English, from Chapter XI in [61] (pp. 153–156).

### 4.5.2. On Abrupt Seasonal changes of the General Circulation

While abrupt seasonal changes of the GCA had been observed over the southern Asian monsoon regions and the Middle East, Yeh and his colleagues extended their studies to East Asia and other regions of the Northern Hemisphere. They made a significant discovery, which they explicitly stated in the *Rossby Memorial Volume*: that the abrupt seasonal change could well be a global phenomenon [62]. Moreover, Yeh and his colleagues conjectured that a certain type of '*instability*' in the atmosphere could be the underlying mechanism:

> "*Concerning the cause of the abrupt changes, we shall only propose the following reasons as a conjecture: From winter to summer the inclination of the sun over the Northern Hemisphere gradually increases. With this increase the temperature contrast between the equator and pole gradually decreases. When it has decreased to a certain value, a certain type of 'instability' in the atmosphere appears and the abrupt change of the upper-air circulation takes place. From summer to winter the reversed sequence of events would occur.*" [62]

Yeh further suggested a possible model experiment "to answer conclusively the above conjecture": "In a rotating half sphere or two coaxial cylinders [. . .] we heat differently the inner and outer part, then gradually decrease or increase the heating difference and observe whether we get the abrupt transition of the circulation as observed in the atmosphere.".

It is remarkable that they took such a broad perspective on abrupt seasonal changes of the GCA during a time when data were limited and model experiments were still in their very early stages. Notably, their conjecture did not mention topography or land–sea contrasts. Meanwhile, Yeh emphasized the central role of large-scale eddies in the maintenance of the GCA (Figure 4); therefore, it is very likely that Yeh and his colleagues believed that eddies would play a part in the speculated instability of the GCA. Interestingly, only half a century later, similarly idealized experiments using general circulation models (GCMs) were conducted, replacing the rotating half sphere or two coaxial cylinders used

in Yeh and colleagues' thought the experiment [69,70]. These GCM experiments revealed that the abrupt seasonal change is caused by the rapid transition between two different circulation regimes: the equinox regime controlled by eddy momentum transport, and the monsoon regime driven directly by thermal forcing. If we interpret the "instability" mentioned in Yeh and his colleagues' conjecture as the instability of circulation regimes, then Yeh's idea aligns qualitatively with the recent findings in [69,70]. For a more comprehensive perspective on the evolution of the theory regarding abrupt seasonal changes in the GCA, refer to [65].

## 5. Period IV: The Period of Transition (1972–1983)

In 1966, the Institute of Atmospheric Physics (IAP) was established from the meteorological section of the IGM, and Yeh's research was stopped for several years due to the Cultural Revolution. It was not until 1972 that Yeh resumed his research activities, and his first project was the study of the GCA through the construction of a laboratory for rotating fluid physics. The significant achievements of these experiments were discussed in the section on Qinghai–Tibetan Plateau Meteorology (Section 4.2). During the period from 1972 to 1983, China underwent a transition from the turmoil of the Cultural Revolution to the era of Reform and Opening-up. Yeh became the director of the IAP in 1978 and later the vice-president of the CAS in 1981. With these roles, he took on more responsibilities in leading scientific research and promoting international cooperation in China [5]. Simultaneously, there was also a transition in the focus of Yeh's investigations from atmospheric dynamics and general circulation to climate dynamics.

During this period, Yeh maintained his theoretical interests in atmospheric dynamics, such as in his investigation of the multiple time-scale theory of atmospheric motions, conducted collaboratively with his former student M.-T. Li [71]. As a collaborator with younger theoreticians at the IAP, Yeh was supportive of the development of Q. C. Zeng's rotational adaptation (adjustment) theory, which focused on the adjustment of atmospheric motions slower than the geostrophic adjustment [72]. Additionally, Yeh contributed to J. P. Chao's spiral-like planetary wave theory in a barotropic atmosphere [73]. Furthermore, he continued to lead the research into Qinghai–Tibetan Plateau Meteorology in China [43].

During the late 1970s and onwards, Yeh's primary focus shifted from atmospheric dynamics towards the field of climate dynamics. This transition coincided with the emergence of global climate models in the 1970s; the publication of the *Charney Report* in 1979 [74]; and the establishment of the World Climate Research Programme (WCRP) in 1980, of whose joint scientific committee (JSC) Yeh became a member from 1982 to 1988 [3].

In 1981, during his visit to the Geophysical Fluid Dynamics Laboratory (GFDL) in Princeton, Yeh collaborated with S. Manabe and R. T. Wetherald. Together, they conducted idealized GCM simulations to investigate the short-term climate effects of snow-cover removal and irrigation [75,76]. Their findings revealed that soil moisture anomalies resulting from these land surface modifications can induce climate anomalies that persist for several months, creating a cross-seasonal "memory" in the general circulation and climate. It is worth noting that Yeh had already gained familiarity with surface and atmospheric energy balance analysis through his earlier work on the northeast trade wind and his subsequent studies on Qinghai–Tibetan Plateau Meteorology and the dynamics of the global circulation as an internally consistent whole. Therefore, it is not surprising to find a unique clarity in the physical insights derived from Yeh and his collaborators' analyses of the surface energy balance and the response of the global circulation to surface anomalies.

## 6. Period V: The Period of Global Change (1984–2013)

During the early 1980s, the international scientific community recognized the importance of the interaction between global biogeochemical processes and physical processes in shaping global environmental and climate change. In response to this realization, the International Geosphere-Biosphere Programme (IGBP) was launched in 1988, symbolizing the establishment of global change science, also known as Earth system science [77].

The concept of "global change" encompasses a wide spectrum of transformations in the Earth's environment, spanning changes in the solid earth, oceans, atmosphere, biosphere, cryosphere, and more [77]. Yeh played a key role from 1984 as one of the pioneering figures who led the launch of the IGBP [78], and he is regarded as the founder of global change studies in China. His ideas, such as those on the sensitive zones of global change [3], were reflected in the early projects of the IGBP. For a more comprehensive account of Yeh's involvement in the establishment of international global change studies, interested readers may refer to [3,79]. However, in this discussion, the focus is only placed on two aspects of Yeh's contributions: the inherent connection between adaptation to global change and the principles of sustainable development, and the concept of "orderly human activities".

At the turn of the 21st century, Yeh recognized the imperative of establishing a strong connection between adaptation to global change and sustainable development. In this context, adaptation refers to the necessary adjustments in natural and human systems to effectively respond to anticipated changes and their resulting impacts, with the aim of minimizing harm and capitalizing on beneficial opportunities [4]. Yeh believed that the lack of awareness among policymakers and the general public regarding global change could impede the success of sustainable development in China. Therefore, he organized a conference in Beijing to discuss how to link sustainable development and the adaptation to global change, inviting about 30 leading scientists of various fields from China, and Yeh's thoughts on the issue were further distilled in [80] and in an interview featured in the *WMO Bulletin* [4].

Yeh conveyed his views in [4,80] regarding the close connection between adaptation to global change and sustainable development. He claimed that the adaptation to global change must align with the principles of sustainable development. Otherwise, the adaptation for temporary or local interests may only result in greater destructive change. On the other hand, sustainable development will not achieve its goals without taking future global change into account. Yeh also analyzed the systematic nature of both sustainable development and adaptation to global change. Yeh emphasized that adaptation to global change must transcend the boundaries between regions, organizations, and business sectors; meanwhile, the trend towards the integration of regional and global economies concisely exemplifies the systematic nature of sustainable development [79].

Yeh and his colleagues further put forward the idea of "orderly human activities", which were defined as human activities that could ensure the maintenance of the life-supporting environment as a whole without notable degeneration, or even with some improvement, while meeting the demands of socio-economic development [81]. It is evident that sustainable development serves not only as the objective of orderly human activities but also as the criterion by which the orderliness of large-scale human activities is measured [3]. Yeh and his colleagues analyzed the attributes of orderly human activities, highlighting their alignment with sustainable development, hierarchical structure, systematic nature, and scale effect [81]. They further suggested an approach to investigating orderly human activities in which scientists, policymakers, and stakeholders at different levels should be closely integrated.

From a retrospective viewpoint twenty years since the publication of [80,81], one can easily identify the visionary nature of Yeh's thoughts on global change, sustainable development, and orderly human activities.

## 7. Conclusions

In this review, Prof. Duzheng Ye's (Tu-cheng Yeh's) research career was divided into five periods, namely: (I) the preparatory period (1935–1944); (II) the Chicago period (1945–1950); (III) the period of breaking ground (1950–1966); (IV) the period of transition (1972–1983); and (V) the period of global change (1984–2013). Figure 5 summarizes the timeline of Yeh's research career with the main foci and selected key publications for each period included. The evolution of his main achievements is provided based on a meta-analysis of his published works. Some of the evaluations of his work by his

contemporaries are provided, and the influences of his achievements are reflected in the literature mentioned here, which represents the advances in the relevant research fields.

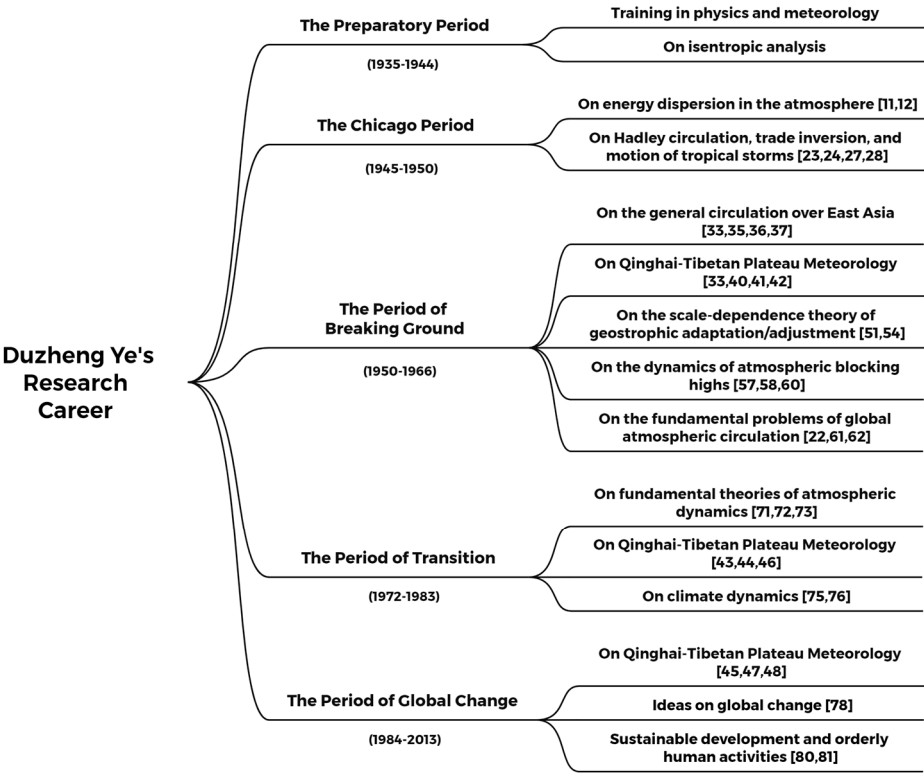

**Figure 5.** The timeline of Yeh's research career and the main foci and selected key publications for each period.

Needless to say, his achievements after he returned to China in 1950 were not only his, but also a part of 'many significant realistic contributions' [32] by the Chinese meteorological community at large [30,82]. In the long course, Yeh played a central role in the explosive development of atmospheric dynamics in China during the 1950s and 1960s, as well as in climate change and global change science in China after the late 1970s. Yeh's dedicated efforts and leadership in the field of global change led to the establishment of a nationwide and decade-long global change research program in China, and he also provided many science-based suggestions on climate change policy to the leaders of China.

It is emphasized that many of Yeh's ideas were very prospective, and hence this review aimed to not only honor past achievements, but also inspire more new scientific findings based on Yeh's visionary ideas. Indeed, Yeh's investigations on energy dispersion in the atmosphere [12] and on the physical mechanism of atmospheric blocking [57,60] are highly relevant to current research endeavors on the possible changes in Rossby wave propagation [83–85] under global warming and the mechanisms of blocking and extreme events [21,86–88]. His study on trade inversion in collaboration with Riehl et al. [24] remains to be used as a mechanistic basis for very recent theoretical and modeling investigations on the dynamics of trade cumuli [25,26]. Yeh's proposal regarding the cause of abrupt seasonal changes of the GCA [62] have been echoed by recent theoretical studies [69,70], and his perspective on the atmospheric general circulation as an internally consistent whole [61] remains the goal of theoretical efforts seeking a unified theory of the GCA [65]. Recent progress on the global effects of the Qinghai–Tibetan Plateau can be found in [89,90].

While many of Yeh's investigations were theory-oriented, Yeh's theoretical investigations were firmly rooted in observations. The availability of observations was indeed very limited in his time, such as during his investigations on the upper-tropospheric circulation over China during the winter of 1945–46 [33], trade inversion [24], and the thermal effects

of the Tibetan Plateau [41]. However, the combination of his physical intuition and insights based on these limited data often led to groundbreaking new findings.

Finally, it should be noted that, due to the space limitations, many of Yeh's scientific contributions were not discussed here, and the emphases in the current review inevitably reflected—and were limited by—the author's understanding of Yeh's works. Suffice it to say, however, many of his scientific ideas are still very relevant to current pivotal issues, such as the need for a universal law of the general circulations of the atmosphere, and human responses to climate change. Therefore, there is no doubt that Prof. Duzheng Ye's works will continue to inspire generations of young scientists.

**Funding:** This research was funded by the National Natural Science Foundation of China, grant number 42042011.

**Acknowledgments:** The author appreciates the assistance in typing the references and reading the manuscript provided by his graduate students Tao Wang, Zhixiang Li, and Yue Gao, and also the assistance of Yue Gao in designing Figure 5. The suggestions from Weijiang Yeh and Weijian Yeh are also appreciated.

**Conflicts of Interest:** The author declares no conflict of interest.

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
