# Peer review of "From General Circulation to Global Change: The Evolution, Achievements, and Influences of Duzheng Ye’s Scientific Research"

_atmosphere, doi:10.3390/atmos14081202_

Round 1

Reviewer 1 Report

This is a nice piece and bringing a new perspective to science by providing historical bare ground of the scientific voyage of Duzheng Ye. It is great to read the chronicle of his scientific exploit.

I have no suggestion for improvement for now

Author Response

I appreciate the positive evaluation of the manuscript from the reviewer. I would further improve the manuscript in the revised version of the manuscript by taking the suggestions from all of the three referees into consideration. Indeed, I hope that more and more scientists of younger generations over the world would better understand and appreciate Prof. Yeh's science at such an era of climate crisis and other crises threatening the very existence of human civilization.  

Reviewer 2 Report

Thank you very much for write so comprehensive due.

This is well review of the scientist, who deserve so nice paper, valuable for practice and science.

Notes: 1. The introduction provided sufficient and good background and include all relevant and enought references. 2. The paper descibe detaily, in five parts of long life periods of researcher by appropriate, in time actual and well details in each era of work prof. Yeh. 3. The methods of presenting mentioned eras are detailed and cleared described. 4. In each fruthfull era author descibe main Professor Yeh results clearly, by text and archive documents. 5. These well-known professor's Yeh achievements included the theory of energy dispersion in the atmosphere, the general circulation of the atmosphere over East Asia and the globe, Qinghai-Tibetan Plateau Meteorology, the scale-dependence theory of geostrophic adaptation (adjustment), and his pioneering ideas on global change. Special emphases are put on some of prof. Yeh’s investigations well ahead of his time, such as his investigations on trade in version, on the general circulation of the atmosphere (GCA) as an internally consistent whole and on abrupt seasonal changes of the GCA, on the global effect of the Qinghai-Tibetan Plateau, on the physical mechanism of atmospheric blocking, and on orderly human activities. 6. One of very important think in professor Yeh work is that we may easily find the visionary nature of Yeh’s thoughts on global change, sustainable development and orderly human activities. 7. Conclusions were supported by the described results and well corelated with them.

Kind Regards.   

Author Response

I appreciate the encouraging evaluation of the manuscript from the reviewer. I would further improve the manuscript in the revision by taking the suggestions from all of the three referees into consideration. Indeed, it is also my hope that more and more scientists of younger generations over the world would better understand and appreciate Prof. Yeh's science at such an era of climate crisis and other crises threatening the very existence of human civilization.  

Reviewer 3 Report

This review paper provides an insightful overview of the research career and contributions of Duzheng Ye, a prominent figure in meteorology and global change science. Overall, the review paper provides a comprehensive overview of Duzheng Ye's research career, and with a few additional details and clarifications, it has the potential to further enhance the reader's understanding of his contributions and their impact on meteorology and global change science. I would suggest a minor revision at this point. The following comments are for the authors' consideration. Followings are the comments and suggestions:

1.  Consider providing a brief overview of the importance and impact of Duzheng Ye's research in the abstract. This will help readers understand the significance of his contributions right from the beginning.

2. Include more specific details about each period to provide a deeper understanding of the key developments and milestones in Duzheng Ye's research. This could include notable publications, collaborations, or major breakthroughs during each period.

3. Provide more examples or specific details about these ahead-of-their-time investigations to illustrate their significance and impact. This will help readers appreciate the innovative nature of Duzheng Ye's work.

4. To further enhance the conclusion, consider including a brief discussion on the current state of research in the areas that Duzheng Ye contributed to and identify potential avenues for further exploration or advancements. This can encourage readers to delve deeper into these topics.

no

Author Response

Response to Reviewer 3:

I appreciate the insightful comments and suggestions from the reviewer very much. And the point-to-point responses are as follows.

This review paper provides an insightful overview of the research career and contributions of Duzheng Ye, a prominent figure in meteorology and global change science. Overall, the review paper provides a comprehensive overview of Duzheng Ye's research career, and with a few additional details and clarifications, it has the potential to further enhance the reader's understanding of his contributions and their impact on meteorology and global change science. I would suggest a minor revision at this point. The following comments are for the authors' consideration. Followings are the comments and suggestions:

  1.  Consider providing a brief overview of the importance and impact of Duzheng Ye's research in the abstract. This will help readers understand the significance of his contributions right from the beginning.

Response: Done as suggested. It is added in the abstract: “His achievements have been central to the development of atmospheric and climate dynamics and global change studies in China, and many of them remain to be fundamental in the context of global climate change.” 

2. Include more specific details about each period to provide a deeper understanding of the key developments and milestones in Duzheng Ye's research. This could include notable publications, collaborations, or major breakthroughs during each period.

Response: Thank you for your suggestion. In preparing the manuscript, I’ve read thorough most of the about 200 papers and the main monographs by Prof. Ye. About 31 key papers (all of them were included two-volume Selected Papers approved by Prof. Ye himself) and 4 monographs were mentioned and analyzed in the manuscript. By following your suggestion, A new figure (Figure 5) is added to highlight the points you mentioned (also attached at the end of the response).

3. Provide more examples or specific details about these ahead-of-their-time investigations to illustrate their significance and impact. This will help readers appreciate the innovative nature of Duzheng Ye's work.

Response: Indeed it was also my initial consideration for the review and I’d added about six figures adapted from Prof. Ye’s publications to provide more details. At last they were not included, not only for the time needed for the copyright issue, but also for that (1) more detailed analysis on specific theoretical issue will be included future publication which is under preparation; (2) my previous papers on “fundamental problems of the GCA” and on abrupt seasonal change of the GCA (with Tapio Schneider) and on global change and sustainable development (in Cultures of Science) have provided in-depth analyses on Prof. Ye’s contributions in these issues; (3) the balance of length of the manuscript for each period of Prof. Ye’s research career.  But I totally agree with your suggestion.  

4. To further enhance the conclusion, consider including a brief discussion on the current state of research in the areas that Duzheng Ye contributed to and identify potential avenues for further exploration or advancements. This can encourage readers to delve deeper into these topics.

Response: Done as suggested. More discussions and 10 more references are added in the final section to address your suggestion. 

Please also see the attachment
